# Dual Functionalized Liposomes for Selective Delivery of Poorly Soluble Drugs to Inflamed Brain Regions

**DOI:** 10.3390/pharmaceutics14112402

**Published:** 2022-11-07

**Authors:** Sabrina Giofrè, Antonio Renda, Silvia Sesana, Beatrice Formicola, Barbara Vergani, Biagio Eugenio Leone, Vanna Denti, Giuseppe Paglia, Serena Groppuso, Valentina Romeo, Luca Muzio, Andrea Balboni, Andrea Menegon, Antonia Antoniou, Arianna Amenta, Daniele Passarella, Pierfausto Seneci, Sara Pellegrino, Francesca Re

**Affiliations:** 1Dipartimento di Chimica, Università degli Studi di Milano, 20133 Milan, Italy; 2School of Medicine and Surgery, University of Milano-Bicocca, 20854 Vedano al Lambro, Italy; 3San Raffaele Scientific Institute, INSPE-Institute of Experimental Neurology, 20132 Milan, Italy; 4San Raffaele Scientific Institute, Experimental Imaging Centre, 20132 Milan, Italy; 5Dipartimento di Scienze farmaceutiche, DISFARM, Università degli Studi di Milano, 20133 Milan, Italy

**Keywords:** matrix metalloproteinases, lipopeptides, blood–brain barrier, neuroinflammation, glibenclamide

## Abstract

Dual functionalized liposomes were developed to cross the blood–brain barrier (BBB) and to release their cargo in a pathological matrix metalloproteinase (MMP)-rich microenvironment. Liposomes were surface-functionalized with a modified peptide deriving from the receptor-binding domain of apolipoprotein E (mApoE), known to promote cargo delivery to the brain across the BBB in vitro and in vivo; and with an MMP-sensitive moiety for an MMP-triggered drug release. Different MMP-sensitive peptides were functionalized at both ends with hydrophobic stearate tails to yield MMP-sensitive lipopeptides (MSLPs), which were assembled into mApoE liposomes. The resulting bi-functional liposomes (i) displayed a < 180 nm diameter with a negative ζ-potential; (ii) were able to cross an in vitro BBB model with an endothelial permeability of 3 ± 1 × 10^−5^ cm/min; (iii) when exposed to functional MMP2 or 9, efficiently released an encapsulated fluorescein dye; (iv) showed high biocompatibility when tested in neuronal cultures; and (v) when loaded with glibenclamide, a drug candidate with poor aqueous solubility, reduced the release of proinflammatory cytokines from activated microglial cells.

## 1. Introduction

Drug candidates often diffuse and distribute freely throughout the body, sometimes resulting in undesirable side effects and insufficient delivery at their site of action, limiting therapeutic efficacy [1]. This is particularly true for central nervous system (CNS) indications, due to the presence of the blood–brain barrier (BBB) [2]. Nanoparticle-based drug delivery platforms have recently emerged as suitable vehicles to overcome pharmacokinetic limitations and non-specific toxicity associated with conventional drug formulations [3].

Acute activation of microglia protects neurons and avoids further tissue damage in stroke and neurotrauma. Conversely, a sustained or chronic pro-inflammatory activation of microglia associates with the release of multiple inflammatory cues that in turn affect CNS homeostasis, contributing to persistent neuroinflammation and leading to tissue damage in chronic neurodegeneration [4]. So called “disease associated microglia” [5] play a role in Multiple Sclerosis (MS), Amyotrophic Lateral Sclerosis (ALS), Alzheimer’s (AD) and Parkinson’s disease (PD), as shown by single-cell RNA analyses of CNS immune cells in neurodegenerative conditions; thus, their modulation is an emerging target against neurodegeneration. 

Glibenclamide, a drug currently used to treat type 2 diabetes, ameliorates neuroinflammation and improves neurological functions [6]. Glibenclamide efficiently inhibits the sulfonylurea receptor 1 (Sur1) [7], that assembles in both the Sur1-Trpm4 and Sur1-Kir6.2 ion channels [8]. Glibenclamide exerts protective effects in several CNS disorders including subarachnoid haemorrhage, traumatic brain injury, ischemic stroke, and status epilepticus [9,10,11,12,13]. Furthermore, glibenclamide-mediated inhibition of Sur1 can dampen pro inflammatory mediators such as tumor necrosis factor (TNF) α, interleukin (IL)-6, and NF-kB in models mimicking cardiac arrest [14]. Because of this large body of literature indicating protective effects of glibenclamide, we selected it as a cargo to load our innovative nanoparticles. 

The limited aqueous solubility of glibenclamide does not prevent its use against BBB-affecting acute CNS injuries [7], but a functional BBB prevents its accumulation in CNS in therapeutically effective concentrations [15]. Additionally, glibenclamide given intracranioventricularly is rapidly removed across the BBB [16]. Effective nanoformulations of glibenclamide for type 2 diabetes treatment were reported recently [17], but neuroinflammation treatment by CNS-permeable nanoparticles is currently unprecedented. Thus, in this study, dual-ligand liposomes were developed for CNS delivery and the controlled release of glibenclamide, as a model of poorly CNS-bioavailable drugs.

Liposomes provide synthetic flexibility, biodegradation, biocompatibility, low immunogenicity and toxicity [18]. Thus, at least fourteen liposomal formulations have been approved by the Food and Drug Administration (FDA), mostly as anticancer agents but also against wet macular degeneration and fungal infections [19]. Recently, functionalization of the liposome surface with ligands or antibodies recognized by specific receptors improved the efficacy of several drugs, conferring them active targeting selectivity [20,21]. Controlled release features could also be achieved through chemically modified components that respond to internal [22] or external stimuli [23]. Accordingly, payload release promoted by a pathological stimulus could provide improved therapeutic efficacy, a better side effect profile, and more opportunities for the safe clinical application in CNS pathologies [24]. In particular, matrix metalloproteinases (MMPs) are a family of zinc-dependent proteases that are upregulated in inflamed CNS microenvironments by activated microglia [25]. 

Therefore, we targeted the pathological up-regulation of MMPs to develop stimulus-responsive liposomes able to in situ release their drug cargo. In particular, dual ligand-modified multitasked nanocarriers developed here contain both a modified peptide derived from apolipoprotein E and MMP-responsive lipopeptides, respectively, to cross the BBB and to release low CNS-permeable, poorly soluble glibenclamide in pathological, inflamed brain regions (Figure 1).

Given their easily scalable production and their proven ability to significantly increase brain access for poorly bioavailable drugs, our multifunctional liposomes represent a versatile platform to address multiple brain diseases with drug candidates suffering from poor pharmacokinetic properties. 

## 2. Materials and Methods

### 2.1. Synthesis and Purification of MSLPs (MMP-Sensitive Lipopeptides)

Fmoc-Lys(stearate)-OH was synthetized as described in Appendix A, according to the literature [26]. Lipopeptides SKK(stearate)SGPLGIAGQSK(stearate)KS (MSLP-1), SKK(stearate)SGAKPRA-LTASK(stearate)KS (MSLP-2) and SKK(stearate)GSALGQIGPSK(stearate)KS (c-MSLP) were prepared by microwave (MW)-assisted Fmoc solid-phase peptide synthesis (Fmoc-SPPS). MSLPs were synthetized on Wang resin (loading between 0.2–0.4 mmol/g), using a CEM Liberty Blue synthesizer. Coupling reactions were performed with a 5 eq. excess of an amino acid (0.2 M in dimethylformamide, DMF), using diisopropyl carbodiimide (DIC, 0.5 M in DMF) and Oxyma Pure (1M in DMF) as coupling reagents. The MW synthesis cycle entailed 15 s (15 s) at 75 °C–170 W, followed by 110s at 90 °C–40 W. In the case of arginine R residues, a double coupling protocol was performed first at room temperature for 25 min, then at 75 °C–30 W for 2 min. N-Fmoc deprotection was performed using 20% piperidine in DMF with a MW cycle entailing 15s at 75 °C–155 W, followed by 60 s at 90 °C–50 W. Final cleavage in solution was obtained by shaking the resin for 3 h in a 90:2.5:2.5:5 TFA/TIPS/H_2_O/phenol mixture. MSLPs were purified in reverse-phase chromatography with a Biotage Isolera instrument equipped with a C18 column. 0.1% trifluoroacetic acid (TFA) in water was used as phase A and 1% TFA in ACN (acetonitrile) as phase B, going from 20 to 100% phase B in 18 min. With such program, we recorded a retention time (Rt) = 16.6 min for MSLP-1; an Rt = 13.5 min for MSLP-2; and an Rt = 15.8 min for c-MSLP. MSLPs were analysed by electrospray ionization high-resolution time-of-flight mass spectrometry (ESI-HR TOF-MS). ESI-HR TOF-MS spectra were recorded on a Q-TOF Synapt G2-Si and values were reported in *m*/*z*. In details, MSLP-1, *m*/*z* 1503.7206 [M + 2H]2+, 702.8170 [M + 3H]3+; MSLP-2, *m*/*z* 2278.5664 [M + H]^+^,1139.7866 [M + 2H]^2+^, 760.1946 [M + 3H]^3+^, 570.3979 [M + 4H]^4+^; and c-MSLP, *m*/*z* 1503.7227 [M + 2H]^2+^, 702.8188 [M + 3H]^3+^. High performance liquid chromatography (HPLC) traces and the evaluation of MSLPs’ structure by circular dichroism (CD) were performed as described in Appendix A.

### 2.2. MMP2 Activity Assay on MSLPs 

Recombinant human MMP2 (rhMMP2, R&D system, Minneapolis, MN, USA) was diluted to 0.2 μg/mL in 200 µL assay buffer (50 mm Tris, 10 mM CaCl_2_, 150 mm NaCl, 0.05% (*w*/*v*) Brij 35, pH 7.5). The enzymatic reaction was started by adding 200 μL of 200 μM MSLPs in the assay buffer, and was stirred at 37 °C for 48 h. Final concentrations were 0.1 μg/mL for MMP2 and 100 μM for MSLPs. The reaction was monitored by HPLC between 1 and 48 h on a small aliquot, by deactivating the enzyme with a 10% aqueous TFA solution. MSLPs’ solutions in assay buffer (see below) were used as a blank. A pronase/unspecific protease activity assay on MSLPs was carried out by diluting a mixture of proteolytic enzymes at 2 ng/mL, 0.2 µg/mL and 20 µg/mL in 200 µL assay buffer (50 mm Tris, 10 mm CaCl_2_, 150 mm NaCl, 0.05% (*w*/*v*) Brij 35, pH 7.5). The enzymatic reaction was started by adding 200 μL of 200 μM MSLPs in the assay buffer, and was stirred at 37 °C for 48 h. Final concentrations of 1 ng/mL, 0.1 μg/mL and 10 μg/mL (Pronase) and 100 μM (MSLPs) were used. The reaction was monitored by HPLC between 1 and 48 h on a small aliquot, by deactivating the enzymes with a 10% aqueous TFA solution. MSLPs’ solutions in assay buffer were used as a blank.

### 2.3. Preparation and Characterization of Liposomes 

#### 2.3.1. Preparation and Characterization of MSLP-Liposomes Functionalized with mApoE Peptide

Conventional liposomes were prepared by thin film hydration followed by probe sonication. Briefly, cholesterol (Chol), sphingomyelin (Sm), and 1,2-distearoyl-sn-glycero-3-phosphoethanolamine-N-[maleimide(polyethylene glycol)-2000] (DSPE-PEG2000-mal, Avanti Polar Lipids, Inc., Alabaster, AL, USA), respectively, in a 48.75:48.75:2.5 molar ratio, were dissolved in chloroform in a round-bottom flask and mixed for 5 min. Then a phospholipid film was obtained through evaporation with a rotary evaporator. The resulting dried lipid film was hydrated for about 2 h by addition of phosphate buffered saline (PBS, pH 7.4) above the lipid phase transition temperature (55 °C). After hydration, unilamellar liposomes were obtained by probe sonication in a continuous mode for 20 min with 30% power delivery (probe type ultrasound Sonics & Materials, Inc., Newtown CT, USA) [27]. To prepare the MSLP-liposomes, MSLP-1, MSLP-2 or c-MSLP were dissolved in methanol at 0.2 mg/mL, and then added to lipids (Chol, Sm, DSPE-PEG2000-mal) previously dissolved in chloroform and mixed. The total mass of MSLPs molecules and Sm amounted to 1.0 mg, with varying MSLP:Sm ratios (1:10, 1:5, 1:1, mass/mass). Solvents were removed by rotary evaporation and liposomes were prepared as described above. In order to target and cross the BBB, MSLP-liposomes were surface-decorated by covalent coupling with a modified peptide derived from apolipoprotein E (mApoE, CWG-LRKLRKRLLR, synthetized by KareBay Biochem, USA) on DSPE-PEG2000-mal, using a 1:10 molar ratio (mApoE:DSPE-PEG2000-mal, mol:mol). The resulting liposomes were purified and characterized as previously described [28,29,30]. Liposomal morphology was characterized by a transmission electron microscope (TEM), using a CM10 Philips instrument (FEI, Eindhoven, the Netherlands) and a negative staining method. Size, polydispersity index (PDI), ζ-Potential and stability over time were analysed by dynamic light scattering (DLS) with an instrument made by Brookhaven Instruments Corporation, Holtsville, NY, USA equipped with a ZetaPALS device, as previously described [28]. 

#### 2.3.2. Preparation and Characterization of Liposomes Loaded with Glibenclamide

MSLP-liposomes were loaded with glibenclamide (Merck Life Sciences, Rome, Italy) as a drug model against neuroinflammation (quantitation and loading assay). Briefly, glibenclamide (0.5 mg) dissolved in methanol (0.13 mg/mL) was added to lipids (5 µmol) dissolved in chloroform (9 mL), either as such or with MSLP-1 (0.2 mg/mL) in methanol at a 1:10 MSLP:Sm ratio (mass/mass). Solvents were removed by rotary evaporation, and liposomes were prepared as described above. The amount of drug embedded in liposomes, as well as the amount released from them, was measured by LC-MS analysis, using an LC/MS 6546 platform composed by an Agilent 1290 II LC system (Agilent Technologies, Palo Alto, CA, USA) coupled to a TOF-MS spectrometer (Agilent Technologies, Palo Alto, CA, USA). Chromatographic separation was achieved using a Zorbax Eclipse Plus C18 column (50 × 2.1 mm, 1.8 μm, Agilent Technologies, Palo Alto, CA, USA). Acetonitrile: 10 mM ammonium formate (60:40 *v*:*v*) containing 0.1% formic acid was used as mobile phase A, with isopropanol as mobile phase B. The elution gradient started with 90% of A for 1.5 min, moving to 10% of A in 7 min and keeping it until 10 min, with 2 min as re-equilibration at 90% of A. The flow rate was 0.25 mL/min, the column temperature was 55 °C, and the injection volume was 1 μL. The MS spectrometer operated in ESI positive mode at a resolving power of 40,000 over a full scan range of 50–1600 *m*/*z*, at a scan rate of 2 spectra/sec with 8 L/min gas flow, 200 °C gas temperature, 35 psi nebulizer, 350 °C sheath gas temperature, 11 L/min sheath gas flow, 2800 Volt capillary voltage, and fragmentator at 160 V. Purine was used as reference mass (*m*/*z* = 121.0509), and continuously infused at 0.08 mL/min flow rate. Two calibration curves were carried out, in order to quantify glibenclamide in liposomes as well as its released amounts. Calibration standards (calibrators) were prepared by spiking liposomes or PBS with a glibenclamide standard solution. Quality control (QC) samples were also prepared by spiking liposomes or PBS with a glibenclamide solution at three concentrations, to validate the analytical method. A quantitative analysis was performed on the extracted ion chromatogram (EIC) at 494.1511 *m*/*z*, corresponding to the protonated ion [M + H]+ of glibenclamide. MS/MS fragmentation spectra of an authentic sample were also used to identify and quantitate glibenclamide. Samples, calibrators and QCs were prepared by adding 600 μL of 9:1 isopropanol/water. After vortex-mixing for 3 min, samples were placed at −80 °C for 1 h and then centrifuged at 3000 rpm for 5 min, and 1 μL of supernatant was injected in the LC-MS system.

### 2.4. Preparation and Characterization of Calcein-Loaded MSLP-Liposomes

In order to assess the retention of payloads in liposomes and their responsiveness to MMPs, calcein was entrapped in the aqueous core of liposomes at a fixed concentration (100 mM) [31]. Briefly, the lipid film containing MSLPs was hydrated with an appropriate volume of a 100 mm solution of calcein (prepared in PBS, pH 7.40), at 55 °C. After complete hydration, unilamellar vesicles were obtained as described earlier. The separation of liposomes from non-encapsulated calcein was achieved by centrifugation (3 spins at 15,000 rpm for 40 min) [32]. Liposomes (20 μL) were diluted with 4 mL PBS, pH 7.40, and fluorescence intensity (FI) was measured (EM 470 nm, EX 520 nm) before and after the addition of Triton X-100 at a final 1%, *v/v* concentration, ensuring liposome disruption and full release of the encapsulated dye. The encapsulated calcein concentration was estimated using a fluorescence intensity (FI) calibration curve for different concentrations of calcein dissolved in PBS (pH 7.4). The stability of MSLP-liposomes was assessed by monitoring cumulative calcein release. Aliquots from the same liposomal suspension where incubated at 37 °C. At different time points, each aliquot was filtered using a 10 kDa MWCO spin filter, at 4000 rpm. The FI of the filtrate, containing calcein released from the liposomes, was measured (EM 470 nm, EX 520 nm). Calcein release was determined by using the following equation: calcein release (%) = [calcein]filtrate/[calcein]initial sample) × 100(1)

The responsiveness of MSLP-liposomes to MMPs was determined by measuring calcein release in presence of activated MMP2 or MMP9. Briefly, recombinant human MMP2 or MMP9 (R&D Systems) were activated using AMPA (amino-phenylmercuric acetate) following manufacturer’s instructions. Then, activated MMPs (50 µL, 0.2 ng/µL for MMP2 or 0.4 ng/µL for MMP9) and MSLP-liposomes (50 µL, around 20 µM of MSLPs) were mixed in a black-well plate. The fluorescence of released calcein was measured with a GENios Microplate Reader (Tecan) in kinetic mode for 10 min. Empty and calcein-loaded chol/Sm/DSPE-PEG2000-mal were used as control samples. Finally, size and PDI were also monitored after incubation with activated MMPs by DLS.

### 2.5. In Vitro BBB Permeability of MSLP-Liposomes

The in vitro BBB permeability model was prepared and characterized, as previously described [30], using human cerebral microvascular endothelial (hCMEC/D3) cells, obtained from Institut National de la Santé et de la Recherche Médical (Inserm, Paris, France) and grown as previously described [33,34,35]. Briefly, cells were seeded (56,000 cells/cm^2^) onto collagen-coated (150 μg/mL rat tail collagen type 1; Gibco, Thermo Fisher Scientific) transwell filters (12-well polyester, 0.4 µm pore size, 1.12 cm^2^ translucent membrane inserts; Costar) to establish a polarized monolayer. The cell monolayer separated into an apical compartment (0.5 mL) representing the blood, and a basolateral one (1 mL) representing the brain. Cells were grown for 3 days in complete EBM-2 medium. Then the medium was replaced with EBM-2 supplemented with 5% FBS, 1% CDLC, 1% P/S, 10 mM HEPES, 5 μg/mL ascorbic acid, 1.4 μM hydrocortisone, and 10 mM LiCl. The integrity of the monolayer was evaluated by measuring transendothelial electrical resistance (TEER), monitored with an STX2 electrode Epithelial Volt-Ohm meter (World Precision Instruments, Sarasota, FL, USA), and by determining the endothelial permeability (EP) to TRITC-dextran 4400 Da (λex = 557 nm, λem = 572 nm) (Sigma Aldrich, Milano, Italy). The formation of tight junctions was evaluated by confocal microscopy (LSM710, Carl Zeiss, Oberkochen, Germany) [36]. At the highest TEER values and lowest EP of TRITC-dextran (day 7–8), MSLP-liposomes functionalized with mApoE were added to the apical compartment (400 nmol of total lipids/well) and incubated at 37 °C for up to 3 h. The concentration and size of intact particles in the basolateral compartment was measured by nanoparticle tracking analysis, using a NanoSight NS300 instrument (Malvern Panalytical Malvern, Cambridge, UK). EP was calculated as described [33,34]. 

### 2.6. Isolation, Culture, and Treatment of Primary Neurons and Cell Lineages

C57BL6J mice were maintained under pathogen-free conditions at San Raffaele Hospital mouse facility (Milan, Italy). Procedures were performed according to the guidelines of the Institutional Animal Care and Use Committee of the San Raffaele Scientific Institute and the Ministry of Health (protocol number 372/2021-PR). Primary neuronal cell cultures were established from C57Bl/6J E17.5 embryonic cerebral cortices (Charles River), according to our published protocol [37]. Briefly, we dissected forebrains in cold HBSS supplemented with 0.6% glucose and 5 mM HEPES pH 7.4 to obtain cerebral cortices. Cortices were mechanically dissociated in single cell preparations and re-suspended in culture medium containing Neurobasal, 5 mM HEPES pH 7.4, 0.6% glucose and the 1× B27 supplement. Cells were plated at 30,000 cells/well on 96 multi wells previously coated with poly lysine (0.2 mg/mL), and primed with Dulbecco’s modified Eagle’s media (DMEM) 20% FBS. Cells were maintained in a humidified 5% CO_2_/95% air environment at 37 °C. Neurons were left to mature for 16 days, and upon visual inspection were used for cytotoxicity assays. 

BV2 microglial cells, RAW264.7 and Neuro2A (N2A) cells were cultured in DMEM supplemented with 10% heat-inactivated FBS, 100 U/mL penicillin and 100 μg/mL streptomycin at 37 °C in a humidified 5% CO_2_ atmosphere. The day before the experiment, BV2 cells were plated on 24 well plates at 80,000 cells/well; LPS was dissolved in PBS as a stock solution (1 mg/mL), and stored at −20 °C. BV2 cells seeded into multiplates were exposed to LPS (0.5 µg/mL) in presence or absence of glibenclamide (40 µM), empty mApoE-MSLP1-liposomes and mApoE-MSLP1 loaded with glibenclamide. Cells were left in vitro for 40 or 60 h, treated with LPS for additional 8 h before using supernatants for an ELISA assessment of TNFα and IL-6.

### 2.7. Extraction of Total RNA

Total RNAs from BV2, RAW264.7 and N2A cells were extracted using the RNeasy Mini Kit (Qiagen, Milano, Italy) according to manufacturer’s recommendations, including DNase (Promega, Milano, Italy) digestion. We synthetized the cDNA using ThermoScript RT-PCR System (Invitrogen Srl, Milan, Italy) and Random Hexamer (Invitrogen), according to the manufacturer’s instructions. RT PCR was done using the following primers for Abcc8, encoding for Sur1: 

F: 5′TTCCCTGGCCACAACCTGCGC3′; 

R: 5′AGACCACAGAGGTGATGGCAGC3′.

Normalization of samples was obtained using the following primers for the housekeeping gene Histone H3: 

F: 5′-GGTGAAGAAACCTCATCGTTACAGGCCTGGTAC-3′; 

R: 5′-CTGCAAAGCACCAATAGCTGCACTCTGGAAGC-3′. 

### 2.8. Cellular Assays Using MLSP-Liposomes

#### 2.8.1. LDH-Glo Cytotoxicity Assay 

A bioluminescent lactate dehydrogenase (LDH)-release assay (LDH-Glo Cytotoxicity Assay, Promega) was used to assess cytotoxicity in supernatants of cells treated with liposomes according to manufacturer’s instruction. Briefly, 2 µL of medium was collected from each well at the end of the treatment, and diluted in the LDH storage buffer (200 mm Tris-HCl (pH 7.3, 10% glycerol, 1% BSA). The LDH storage buffer + medium solution was subsequently mixed in a 1:1 ratio with LDH detection reagent in a white 96-multiwell plate (Corning, NY, USA), and incubated for 30 min at room temperature. Luminescence was measured using the Mithras LB 940 instrument (Berthold technologies, Bad Wildbad, Germany), with an integration time of 0.8 s. Maximum LDH Release Control was obtained lysing untreated cells with 10% Triton X-100 (2 μL, Sigma-Aldrich, Milan, Italy). In each experiment, we calculated the % of cytotoxicity as follows:(Exp.LDH Release − MB)/(Max LDH Release Control − MB)(2)
where MB is Medium Background.

#### 2.8.2. Cell Counting Kit-8 Assay

Cell viability was measured using the Cell Counting Kit-8 (CCK8) Metabolic Assay Kit, following the manufacturer’s protocol (Sigma-Aldrich, Burlington, VT, USA). Briefly, a CCK-8 solution (10 µL) was added to the cell medium (100 µL) in each well at the end of the treatment. Plates were incubated for 1–3 h at 37 °C. We measured samples absorbance every 30 min at 450 nm, using a Multiskan Sky (Thermo fisher scientific, Waltham, MA, USA). Absorbance readings were subsequently normalized to the mean value of the controls.

#### 2.8.3. ELISA Assay 

BV2 cells were seeded in 24-well plates at 80,000 cells/well, then treated with LPS (0.5 µg/mL) to induce TNFα and IL-6 production. Cells were treated for 8 h. Cytokine levels in cells supernatants collected before and after treatment were assayed using the mouse TNF-alpha DuoSet ELISA (R&D Systems, DY410) and the mouse IL-6 DuoSet ELISA kit (R&D Systems, DY406), according to manufacturer’s instructions. 

### 2.9. Statistical Methods

Each experiment was conducted at least in triplicate. We express data as the mean value ± standard deviation (SD) of independent experiments. We assessed the normality of data applying either the Kolmogorov–Smirnov test (with Dallas–Wilkinson–Lille for *p* value) or the D’Agostino and Pearson omnibus normality test. Comparisons were made using either the unpaired t-test, the one-way or two-way analysis of variance (ANOVA) tests, followed by Tukey’s multiple Comparison test or by Bonferroni multiple comparison test. Statistical analyses were performed using PRISM5.01 (GraphPad Software, La Jolla, CA, USA). Values lower than 0.05 were considered statistically significant.

## 3. Results

### 3.1. Synthesis of MSLPs

MMP-sensitive lipopeptides (MSLPs, Figure 2) were prepared as key functional molecules able to assemble into liposomes when mixed with lipids and phospholipids. The fundamental core of MSLPs is represented by MMP2 [38,39] and both MMP2- and MMP9-responsive [40] sequences (GPLGIAGQ and SGAKPRALTA, respectively). In order to increase their hydrophilicity we inserted Ser and Lys, respectively as polar and protonated residues, on both sides of the MMP sequence. Such sequences are surrounded by two hydrophobic tails, obtained by lipidation of Lys residues with stearic acid. The choice of protonated Lys in the linker in order to increase MSLP hydrophilicity was based on the hypothesis that a positive charge carried by MSLPs may facilitate their interaction with negatively charged phospholipids, thus increasing the chance of more stable nanostructures [41,42]. The resulting SKK(stearate)SGPLGIAGQSK(stearate)KS (MSLP-1) and SKK(stearate)SGAKPRA-LTASK(stearate)KS (MSLP-2) sequences (Figure 2a,c) were synthetized in good amounts and purity (>90%) by MW-SPPS (Figure 2b,d and Appendix A). As a negative control, a scrambled MSLP-1 sequence SKK(stearate)GSALGQIGPSK(stearate)KS (c-MSLP, Figure 2e,f) was prepared, characterized by HPLC and also tested.

Circular dichroism (CD) experiments were then carried out to investigate the effect of the lipid bilayer membrane of liposomes on MSLPs’ conformation. MSLPs were analysed both as such (100% TFE and 1:1 TFE/H_2_O) and in presence of 1:1 Sm/Chol (1:10 peptide/lipid ratio). CD spectra (Appendix A) showed that free MSLPs lacks a preferred conformation. However, when lipopeptides were mixed with Sm and Chol, an increase of β-sheet content and a decrease in random coil one were observed (Appendix A). These results suggest that a hydrophobic interaction of MSLPs with lipids enhances the hydrophobicity of MSLPs, partially stabilizing a defined MSLP conformation [43,44]. 

### 3.2. Characterization of MSLP-Liposomes

mApoE-free liposomes, embedding MSLP molecules at different MSLP:Sm ratios (mass/mass), were prepared by lipid film hydration followed by probe sonication, diluted in PBS pH 7.4 and their size was measured by DLS. Liposomes displayed a < 200 nm diameter when prepared at lower MSLP/Sm mass ratio; while an increase of MSLP/Sm mass ratio led to much larger liposomes, reaching a > 500 nm diameter (Figure 3a). For this reason, further experiments were performed using 1:10 or 1:5 MSLP:Sm mass ratios. Liposomal size was then monitored over time by DLS, observing only a slight increase in diameter of MSLP-liposomes after 7 days in PBS at room temperature (Figure 3b). 

To determine the integrity and stability of MSLP-liposomes, the fluorescent small molecule probe calcein was entrapped in the liposome core and its cumulative release was measured at different times (Figure 3c). Up to 72 h, MSLP-1 and MSLP-2 liposomes showed a marginal, <2% calcein release, which slightly increased to ~10% at day 7, and further increased to ~35% after 35 days; such profile is compatible with their in vivo use in pharmacological experiments. Conversely, scrambled c-MSLP liposomes showed a faster and higher release of calcein already detectable at 1 h, reaching ~30% release at day 7, and >60% release after 35 days. Calcein release profiles of liposomes containing different amount of the same MSLP molecule were comparable (Figure 3c). MSLP-liposomes displayed a 0.2 PDI and negative ζ-potentials. MSLP-liposomes loaded with calcein showed comparable, activity-compliant values (Figure 3d).

Surface functionalization of MSLP-liposomes via a covalent thiol-maleimide linkage with a mApoE-derived peptide sequence did not significantly affect the size of liposomes, which remained <200 nm, and the ζ-potential, which remained around—30 mV, accordingly to previous results [29,30]. Assuming that 70,000 lipid molecules are in the outer layer of a liposome with a 140 nm diameter and containing 2.5 mol% of DSPE-PEG2000-mal, and a 70%coupling efficiency, the density after incubation is ∼1000 peptide molecules per single liposome. A representative TEM image of negatively stained MSLP-liposomes is shown in Figure 3e, revealing a densely packed dispersion of aggregates consisting mostly of small, spherical, unilamellar vesicles, with diameters ranging between 100 and 200 nm. Slight deviations from the expected spherical shape of the vesicles, and the visible aggregation observed on TEM grids might be attributed to the combined effect of staining and drying. A TEM image of diluted samples, in which a spherical shape is more visible, is reported in Appendix A.

### 3.3. MMP-Responsiveness of MSLP Constructs and MSLP-Liposomes 

The sensitivity of MSLPs for MMP2 was first assessed by incubating lipopeptides with the enzyme (0.1 μg/mL) for 48 h at 37 °C, monitoring any proteolytic cleavage by HPLC at different times. For all sequences, a partial cleavage of the peptide sequence could be observed after 1 h through a reduction of their HPLC peaks. Nevertheless, while the MSLP-1 peak completely disappeared after 48 h, the peaks relative to MSLP-2 and c-MSLP were still present after 2 days’ incubation (Appendix A), indicating, as expected, a higher sensitivity towards MMP2 for MSLP-1. A control experiment was then carried out using unspecific pronase proteases to exclude off-target cleavage due to unspecific proteolysis. MSLP-2 was incubated with pronase at 1 ng/mL, 0.1 μg/mL and 10 μg/mL concentrations (Appendix A). At lower concentrations, no significant change of the intensity of MSLP-2 peaks was observed, showing good stability towards other proteases. Only in the presence of 10 μg/mL concentrations of pronase, largely exceeding common blood concentration of pepsin (130 ng/mL) [45] and pepsinogen I, a complete cleavage of MSLP-2 was observed. The MMP responsiveness of calcein-loaded MSLP-liposomes was determined by measuring calcein release in presence of activated MMP2 or MMP9 (Figure 4). In presence of activated MMP2, a significant increase of calcein release over 15 min was detected comparing calcein-loaded MSLP-1 (Figure 4a) and MSLP-2 liposomes (Figure 4b) with calcein-loaded scrambled cMSLP liposomes (Figure 4c); please note that the curves were adjusted by subtracting the calcein release values observed from MSLP-free liposomes. The highest calcein release was measured for MSLP-1 liposomes, and for both MSLP-1 and MSLP-2 liposomes with a 1:10 MSLP:Sm ratio (mass:mass) (Figure 4a,b). In particular, Vmax values for 1:10 MSLP:Sm ratios, calculated by Michaelis-Menten least squares fit, were 5.65, 3.81 and 2.53 calcein release/min for MSLP-1 liposomes, MSLP-2 liposomes and c-MSLP liposomes, respectively. A similar trend was observed in presence of activated MMP9 (Figure 4d–f), with the highest calcein release measured for MSLP-1 liposomes with a 1:10 MSLP-1:Sm ratio (mass/mass). The estimated Vmax values for MSLP-1, MSLP-2 and c-MSLP liposomes were 5.38, 4.11, and 1.28, respectively (1:10 MSLP:Sm ratio). Calcein release from MSLP-2 liposomes and c-MSLP liposomes (Figure 4e,f) was <2% and <1%, respectively, with similar values for 1:10 and 1:5 MSLP:Sm ratios. After incubation with activated MMPs, the size and PDI of MSLP-1 liposomes were measured by DLS. The results showed that both these parameters increased to 237.7 ± 23 nm and 0.33 ± 0.02 mV, respectively. 

The release of calcein from MSLP liposomes (1:10 MSLP:Sm, mass/mass) functionalized with a mApoE sequence induced by the presence of activated MMP2 or MMP9 is shown in Figure 5. Once more, the highest release of calcein was detected for MSLP-1 liposomes, the lowest release resulted from c-MSLP liposomes, and a higher calcein release was obtained in presence of activated MMP2 (Figure 5a) in comparison with activated MMP9 (Figure 5b).

The ability of mApoE-MSLP-liposomes to cross the BBB was evaluated in vitro using hCMEC/D3 cells seeded on a transwell system. The BBB model was characterized by measuring its bioelectrical (TEER), structural (presence of junctions by confocal microscopy), and functional properties (EP of a fluorescent probe). TEER values progressively increased over time, reaching ≥ 45 Ω cm^2^ values at 7–8 days after cell seeding (Figure 6a). At the same time, the formation of junctions was evaluated by confocal microscopy and by measuring the paracellular passage of TRITC-dextran (4400 Da). Namely, claudin-5 and VE-cadherin were formed 7 days after seeding in the hCMEC/D3 monolayer (Figure 6b). The EP to TRITC-dextran resulted to be 7.32 ± 0.95 × 10^−5^ cm/min.

In these conditions, mApoE-MSLP-liposomes were added to the apical transwell compartment, and the number of mApoE-MSLP-liposomes in the basolateral compartment was measured up to 3 h by a NanoSight instrument. All tested mApoE-MSLP-liposomes were similarly able to cross the endothelial monolayer (Figure 6c), with an average EP of 3.06 × 10^−5^ cm/min, in comparison to naked liposomes (EP = 5 × 10^−6^ cm/min), suggesting that the presence of MSLP-1 does not alter the BBB crossing performance of liposomes, confirming that the key factor to overcome the BBB is the functionalization with mApoE. Glibenclamide, a selective KATP ion channel modulator, was embedded in liposomes as a poorly BBB-permeable, poorly soluble drug model [15]. LC-MS results showed that the yield of drug incorporation was 70 ± 12%, corresponding to 106 ± 10 µg glibenclamide/µmol of total lipids. DLS analysis showed that MSLP-liposomes embedding glibenclamide displayed a suitable 160 ± 10 nm diameter. The ability of MSLP-liposomes to retain glibenclamide was determined by monitoring drug release from liposomes over time by LC-MS in PBS solution without MMPs. Results showed that drug release was <1% up to 14 days (Appendix A).

### 3.4. Biocompatibility of mApoE-MSLP-Liposomes

We tested cell toxicity elicited by mApoE-MSLP1-liposomes loaded with glibenclamide on primary neurons established from mouse E17.5 forebrains. We used mApoE- liposomes carrying the MSLP-1 core sequence, that provided the highest MMP-driven release of calcein and good glibenclamide loading in previous experiments. Upon plating, primary neurons were left to develop in vitro for 16 days and then treated with DMSO—i.e., the solvent used to dissolve glibenclamide—for 24 h. We observed a slight but significant reduction of cell viability of neurons exposed to DMSO (Figure 7a). We next incubated neurons either with glibenclamide-loaded mApoE-MSLP-1 liposomes, or with increasing concentrations of free glibenclamide, mirroring the concentration of the embedded drug payload in liposomes. Treatment for 24 h did not affect cell viability of neurons when compared with controls receiving DMSO (Figure 7b). Supernatants from these cultures, scored for LDH release, confirmed that both treatments did not increase cytotoxicity in primary neurons (Figure 7c).

### 3.5. Effects on LPS-Induced Cytokine Release by Glibenclamide-Loaded mApoE-MSLP1-Liposomes 

We next setup a cell-based assay to test the efficacy of glibenclamide-loaded mApoE-MSLP1-liposomes. Using RT-PCR we assayed the expression of Sur1 in immortalized BV2 microglia, RAW264.7 macrophages and N2A neuroblasts. Abcc8 encodes for the ATP binding cassette (ABC) of Sur1, and therefore we wanted to assess whether BV2 cells used in our experiments were expressing this transcript. Abcc8 transcripts were detected in BV2 microglia, while RAW264.7 and N2a cells did not express this mRNA (Figure 8).

Glibenclamide is a potent inhibitor of Sur1 in microglia, and it is able to reduce the release of pro-inflammatory cytokines TNFα, IL-6 and IL-1β induced by LPS stimulation [14]. Glibenclamide-loaded mApoE-MSLP-1 liposomes need activated MMP2 to release their content. In vitro activation of recombinant MMP2 requires incubation of MMP2 pro-domain with 1 mM p-aminophenylmercuric acetate (APMA) at 37 °C. We tested potential cytotoxic effects deriving from APMA and/or activated MMP2 on BV2 cells, measuring cell viability with the CCK8 assay and the LDH content assay, without observing any cytotoxic effects in both cases (Appendix A). We next exposed BV2 cells to MMP2-activated, glibenclamide-loaded mApoE-MSLP-1 liposomes before stimulating cells with LPS. Controls were made by incubating BV2 cells with activated MMP2 along with glibenclamide-loaded mApoE liposomes lacking the MSLP lipopeptides. We incubated microglia with both liposomes for 40 h, and then cells received the LPS treatment for a further 8 h. Both TNFα and IL-6 were significantly increased in LPS-stimulated BV2 cells when compared with untreated cells. Levels of both cytokines were partially reduced by glibenclamide-loaded liposomes lacking MMP2 sensitive lipopeptides, albeit a significantly higher reduction of TNFα and IL-6 were obtained in BV2 cells receiving glibenclamide-loaded mApoE-MSLP-1 liposomes (Figure 9).

Since the incomplete control of TNFα release after 40 h incubation, we next incubated BV2 cells with either glibenclamide-loaded mApoE-MSLP-1 liposomes or with MMP2-insensitive glibenclamide-loaded liposomes for 60 h, to assess whether increasing glibenclamide release from liposomes could enhance the immunomodulation of cells. Glibenclamide loaded in MMP2-insensitive liposomes induced a reduction of both TNFα and IL-6 (red, Figure 10a,b), confirming previous observations. However, TNFα levels were barely detectable in cells treated with glibenclamide-loaded mApoE-MSLP-1 liposomes and exposed to MMP2. Similarly, we observed in these cultures a 10-fold reduction of IL-6 levels when compared with cells receiving glibenclamide loaded in MMP2-insensitive liposomes (green, Figure 10a,b). 

We also treated BV2 cells with free glibenclamide (10 to 80 μM, Figure 10c,d) for 60 h before stimulation with LPS, showing a dose-dependent reduction of both TNFα and IL-6 levels. Interestingly, the reduction of TNFα and IL-6 elicited by glibenclamide-loaded mApoE-MSLP-1 liposomes (estimated 40 μm glibenclamide concentration) was higher than the one observed in response to free 80 µm glibenclamide, thus suggesting that the slow payload release provided by glibenclamide-loaded mApoE-MSLP-1 liposomes can increase drug efficacy. We noticed a partial reduction of both cytokines even in MMP2-insensitive, glibenclamide-loaded mApoE-liposomes. These results recapitulate calcein release observed in MSLP-free vs. MSLP-containing liposomes, and suggest an aspecific effect of MMP2 (and maybe other proteolytic enzymes) which will be further studied.

## 4. Discussion

In the last decades, nanoparticles have been used to enhance the efficacy of drugs designed to target the CNS. Increased drug bioavailability, limited undesirable side effects, and an increased accumulation of neuroprotective agents in CNS compartments were obtained [46,47]. In particular, the surface modification of nanoparticles with peptides, small molecules, or antibodies able to cross the BBB and to recognize specific brain disease targets have been exploited to improve the CNS-targeted delivery of nanoformulations [48]. 

We built dual-functional liposomes to deliver poorly bioavailable drugs across the BBB, and to release them in response to up-regulated MMPs in neuroinflammatory microenvironments. Such profile is due to (i) the co-assembly of lipids with MMP-responsive peptide-fatty acid chimeras (MSLP molecules), yielding nano-sized liposomes able to release their payload in response to increased MMP2 and MMP9 levels; and to (ii) the surface functionalization of such liposomes with a modified peptide derived from apolipoprotein E (mApoE), capable, in vitro and in vivo, to mediate brain drug delivery [29,30,49,50,51]. 

Peptide-containing liposomes, undergoing MMP-dependent cleavage to destabilize their membrane and to release their payload in a systemic tumor microenvironment, were reported [38,52]. We compared their properties with our mApoE-containing, MMP-cleavable MSLPs after their synthesis, purification, structural and activity characterization.

We studied the influence of 1:10 and 1:5 MSLP:Sm (mass ratios) in MLSP lipopeptides, similar to literature counterparts [52]. Our MSLP-liposomes prepared in PBS at pH 7.4 displayed a uniform size distribution, with a <200 nm diameter. The latter is suitable for an improved transport of drugs across biological barriers by using small size nanoparticles (preferably < 200 nm) [53]. 

We loaded MSLP-liposomes with calcein to determine their stability; their release profile at 37 °C in PBS was measured by a fluorescence assay. MSLP-1 and MSLP-2 liposomes did not exhibit a significant release of calcein after 7 days (<10%), while scrambled c-MSLP-liposomes were less stable (30% calcein release in 7 days) [52]. Our MSLP-liposomes were more stable in unchallenged conditions than other MMP-sensitive liposomes [52]. Their higher content of both cholesterol and saturated long acyl chain lipids (i.e., sphingomyelin) could possibly reduce the non-specific permeability of MSLP-liposomes and cause a slower, prolonged release rate of entrapped payloads [54]. We underline that our MSLP-liposomes are nanoparticle formulations similar to Marqibo^®^ [55], an FDA-approved liposomal formulation endowed with high in vivo stability. 

MSLP-1 liposomes exhibit a better and faster MMP-dependent calcein release profile than MSLP-2 liposomes, and an increase of size and PDI suggested a MMP-dependent destabilization of liposomes bilayer. Lower 1:10 MSLP:Sm ratios (≈1.8 molar % of MSLP) showed a better MMP-responsive behavior compared to higher, 1:5 MSLP:Sm ratios (≈4 molar %) for MSLP-1 and MSLP-2 liposomes. This is possibly due to a better accessibility for MMPs to liposomes with a ≤2 molar % MLSP content, as shown for MMP-sensitive immunoliposomes with a 1 molar % MMP-sensitive lipid content [38]. Further surface functionalization with an mApoE-derived peptide sequence led to a sigmoidal calcein release profile in presence of activated MMPs, once better and faster for MSLP-1 liposomes. Such a profile should be translatable into a time- and rate-controlled and sustained release capacity for mApoE MSLP-1 liposomes in vivo [56,57,58]; conversely, higher peptide/lipid ratios and faster MMP-induced payload release for MMP-targeted liposomes in literature [52] may be better suited for systemic tumor targeting. It is important to point out that the maximum calcein release from liposomes incubated with activated MMPs is ~6%. Since this result was obtained in vitro with a limited amount of enzymes, it would be more appropriate to evaluate the suitability of these liposomes in a more complex and biologically relevant environment. 

ApoE-derived peptides are known to mediate the delivery of drug payloads in the brain [33,50,51]. Our mApoE MSLP-liposomes were confirmed to cross the BBB by using an in vitro model. Their endothelial permeability was in the same order of magnitude of published data [33,48], suggesting that a low percentage of MSLP molecules inserted into liposomes did not affect their penetration through a biological barrier model.

Finally, to validate our dual-functional liposomes in a biological efficacy model, mApoE MSLP-liposomes loaded with glibenclamide were tested in vitro. Glibenclamide is an antidiabetic drug, recently evaluated in models of acute CNS injuries [7]. It regulates the pro-inflammatory activation of microglia by inhibition of overexpressed, activated Sur1-KATP channels after acute and chronic neuroinflammation [8,9]; improves the course of Experimental Autoimmune Encephalomyelitis (EAE), a multiple sclerosis model, by inhibiting up-regulated Sur1-Trpm4 channels in astrocytes [59]; and blocks NLRP3 inflammasome/IL-1β signalling by down-regulation of the release of pro-inflammatory cytokines and reactive oxygen species. These features would make glibenclamide a prospective tool against neuroinflammatory conditions, but its poor brain bioavailability and aqueous solubility are limiting factors for the treatment of CNS disorders. 

mApoE MSLP-liposomes loaded with glibenclamide showed good loading, encapsulation efficiency and drug retention, in agreement with the encapsulation, stability and drug release of known glibenclamide-loaded liposomes [60]. We tested the tolerability of mApoE-MSLP-1-liposomes loaded with glibenclamide on mouse cortical neuronal cells, showing neither any sign of cytotoxicity nor metabolic alterations. We also demonstrated that glibenclamide-loaded mApoE-MSLP1-liposomes efficiently control the release of TNFα and IL-6 from LPS-activated microglial cells, and that their effect is strongly increased by the presence of an MMP2 sensitive sequence, ensuring a controlled drug release at effective concentrations in the inflamed brain regions.

## 5. Conclusions

Some small molecules aiming for CNS indications share poor brain bioavailability (BBB permeability and/or solubility) and systemic side effects, making them challenging molecules to be administered. Triggered drug release, in association to targeted delivery systems for nanoparticles, offers the possibility to overcome common toxic side effects and bioavailability issues for the treatment of neurological diseases. In the present work, we exploited the over-expression of extracellular MMPs in CNS disease microenvironments, as an enzymatic trigger for drug release in the proximity of diseased cells. MMP-sensitive liposomes carrying neuroprotective therapeutics across the BBB resulted from the co-assembly of lipids with MMP-responsive peptide-fatty acid chimeras (MSLP molecules), and their further surface decoration with a mApoE-derived peptide sequence for BBB crossing. Rationally designed mApoE MSLP-liposomes could cross a BBB-like layer in vitro, and release the poorly bioavailable selective K_ATP_ ion channel modulator glibenclamide, whose activity regulates the release of proinflammatory cytokines from activated microglial cells. Given the easy production and scale up of mApoE-MSLP-liposome batches, their CNS compliance and selective payload release in pathologic conditions, this strategy may provide opportunities to build up clinically compliant nanoobjects to address the complexity of several CNS diseases.

## Figures and Tables

**Figure 1 pharmaceutics-14-02402-f001:**
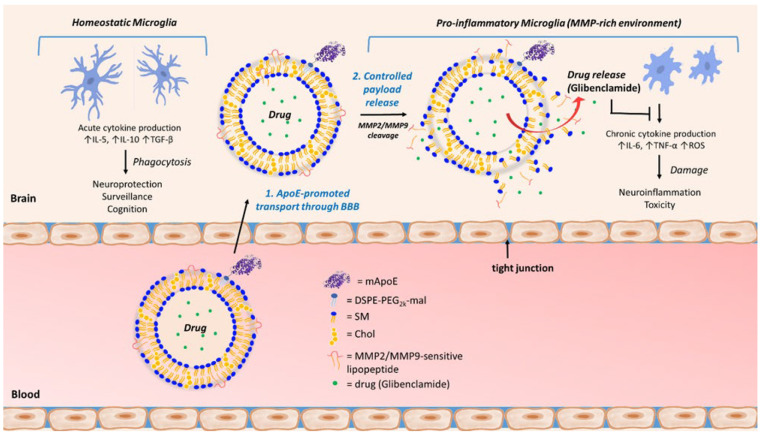
CNS-targeted multifunctional liposomes useful (1) to promote drug brain bioavailability, by mApoE-induced transport through the BBB; and (2) to promote specific and controlled release of loaded, poorly bioavailable neuroprotective drugs by MMPs-sensitive lipopeptides.

**Figure 2 pharmaceutics-14-02402-f002:**
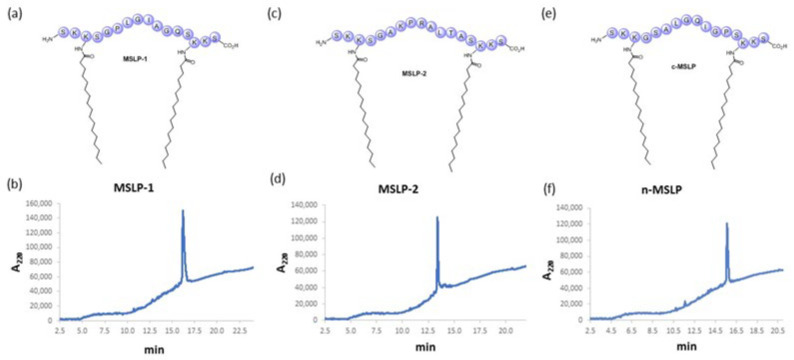
MSLPs molecules, characterized by a C18 alkyl hydrophobic moiety and a peptide core, corresponding to a MMP2- (MSLP-1, (**a**)) and a MMP2/9-responsive sequence (MSLP-2, (**c**)), and to a scrambled sequence (c-MSLP, (**e**)). HPLC chromatograms of the lipopeptides (purity > 90%) are provided respectively in (**b**,**d**,**f**).

**Figure 3 pharmaceutics-14-02402-f003:**
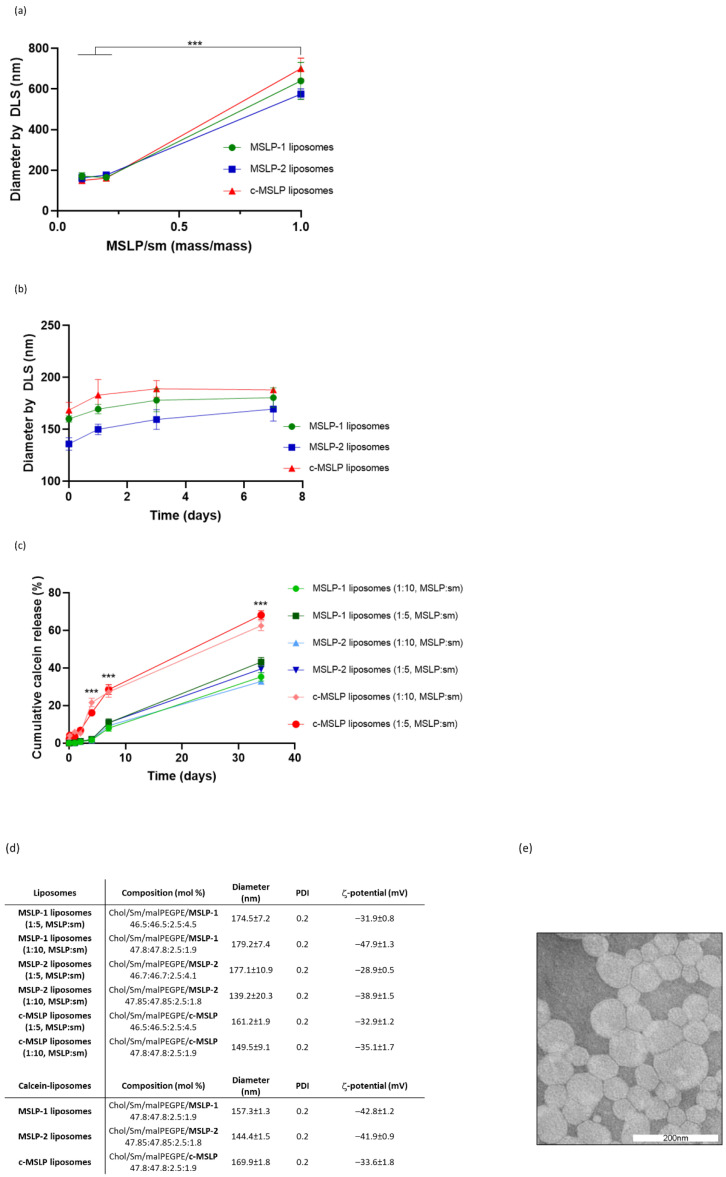
Characterization of MSLP-liposomes. (**a**) Hydrodynamic diameter of MSLP-liposomes prepared with different MSLP/Sm mass ratios, measured by DLS. (**b**) Hydrodynamic diameter of MSLP-liposomes measured by DLS at different time points. Dotted lines, 1:5 MSLP:Sm (mass/mass); continuous lines, 1:10 MSLP:Sm (mass/mass). (**c**) Cumulative calcein release from MSLP-liposomes. Inset: calcein release at short time points. Dotted lines, 1:5 MSLP:Sm (mass/mass); continuous lines, 1:10 MSLP:Sm (mass/mass). (**d**) Diameter, polydispersity index (PDI) and ζ-potential of MSLP-liposomes, empty or embedding calcein, measured by DLS and ZetaPALS device. (**e**) Representative TEM image of MSLP-liposomes (scale bar 200 nm). Results are expressed as mean ± SD (*n* = 10). *** *p* < 0.001 (Student’s *t* test).

**Figure 4 pharmaceutics-14-02402-f004:**
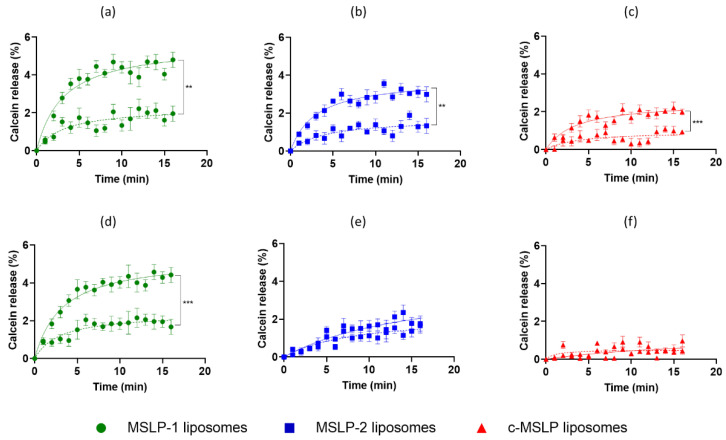
Calcein release from MSLP-liposomes in presence (top traces) or absence (bottom traces) of either activated MMP2 (**a**–**c**) or MMP9 (**d**–**f**). All data were subtracted of calcein release values from calcein-loaded chol/Sm/DSPE-PEG2000mal-liposomes. Results are expressed as mean ± SD. ** *p* < 0.01, *** *p* < 0.001 (Student’s *t* test). One-way ANOVA found a significant effect of treatment (MMP2 groups: F_(5,96)_ = 29, *p* < 0.001; MMP9 groups: F_(5,96)_ = 41, *p* < 0.001). Dotted lines/bottom, 1:5 MSLP:Sm, (mass/mass); continuous line/top, 1:10 MSLP:Sm, (mass/mass).

**Figure 5 pharmaceutics-14-02402-f005:**
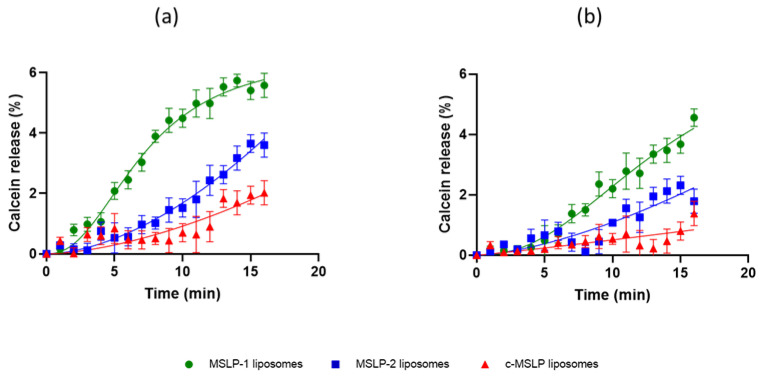
Calcein release from mApoE-MSLP-liposomes (1:10 MSLP:Sm, mass/mass) in presence of activated (**a**) MMP2 or (**b**) MMP9. Curves were subtracted of calcein release values from calcein-loaded chol/Sm/DSPE-PEG2000mal-liposomes. Results are expressed as mean ± SD. One-way ANOVA found a significant effect of treatment (MMP2: F_(5,96)_ = 5.6, *p* = 0.0001; MMP9: F_(5,94)_ = 6.7, *p* < 0.001).

**Figure 6 pharmaceutics-14-02402-f006:**
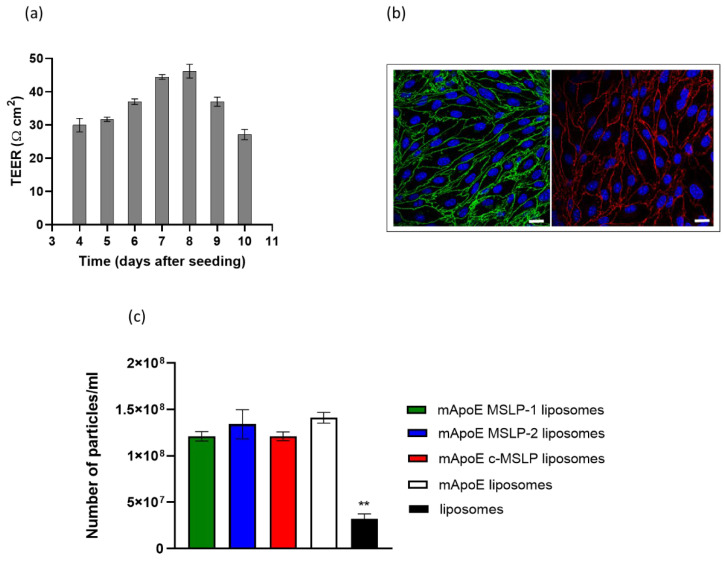
Characterization of a BBB in vitro model and evaluation of its permeability to MSLP-liposomes. (**a**) TEER values of the cell monolayer from day 4 to day 10 of its growth on a transwell. (**b**) Confocal microscopy visualization of junctions (blue, nuclei; green, claudin-5 and red, VE-cadherin). Bars represent 20 μm. (**c**) Evaluation of the permeability of mApoE-MSLP-liposomes across an endothelial monolayer after 3 h incubation by NanoSight. Results are expressed as mean ± SD (*n* = 3). ** *p* < 0.01 (Student’s *t* test).

**Figure 7 pharmaceutics-14-02402-f007:**
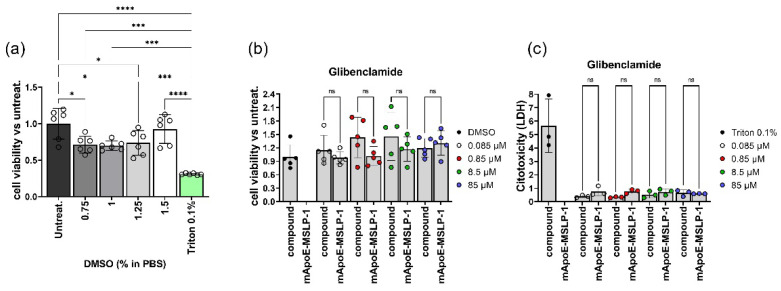
(**a**) Neurons (30 k/well) received increased amount of DMSO (0.75, 1, 1.25 and 1.5%). The histogram shows fold changes over untreated controls (mean values ±SD). Neurons received either mApoE-MSLP-1-liposomes loaded with glibenclamide (0.085, 0.85, 8.5 and 85 µM), or the free drug at the same concentrations for 24 h before supernatants were collected for the CCK8 assay (**b**) and the LDH release assay (**c**). Histograms of panels b and c show fold variation over DMSO-treated cultures (mean values ± SD) for the CCK8 assay and over triton-treated cultures (mean values ± SD) for the LDH release assay. Color codes indicate different glibenclamide concentrations in tested mApoE-MSLP-1-liposomes. Dots show individual experiments; ns not significant, * *p* < 0.05, *** *p* < 0.001, **** *p* < 0.0001. One way ANOVA followed by Tukey’s multiple comparisons test.

**Figure 8 pharmaceutics-14-02402-f008:**
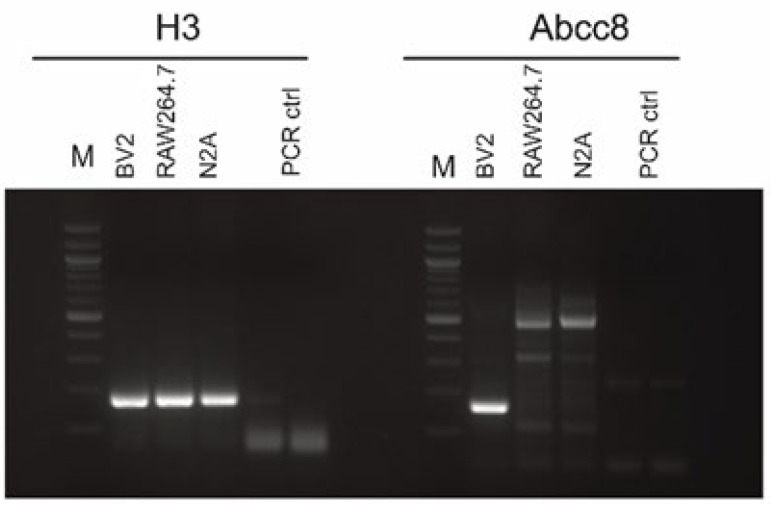
RT-PCR for H3 and Sur1/Abcc8. Five ng of cDNA were used in each PCR to amplify H3 (198 bps) and Abcc8 (166 bps) amplicons. Controls were obtained performing retro transcriptions on water, as well as including water in the PCR as an internal control. M: 100 bp ladder.

**Figure 9 pharmaceutics-14-02402-f009:**
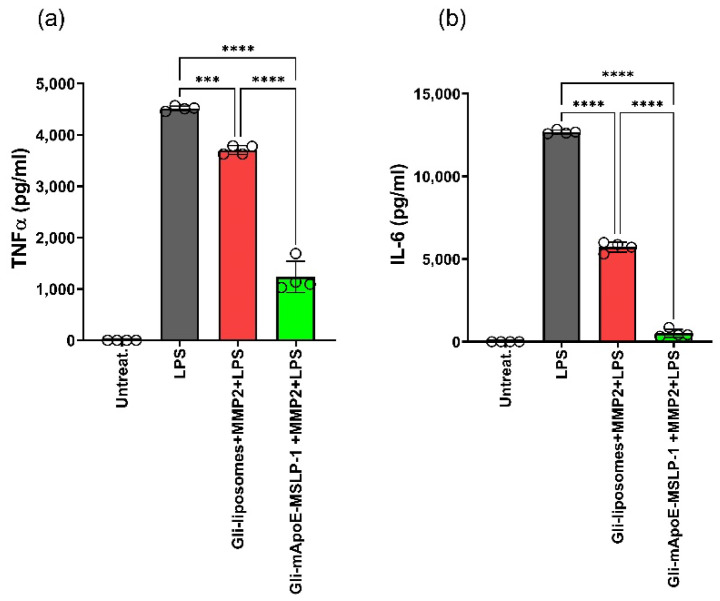
BV2 cells (80 k/well) were exposed to activated MMP2, MMP2-sensitive (green) and insensitive (red) glibenclamide (40 µM)-loaded mApoE liposomes for 40 h. Cells were then stimulated with LPS for 8 h before assaying TNF-α (**a**) and IL-6 levels (**b**) in supernatants. Levels of both cytokines were established by an ELISA assay; dots indicate individual experiments, and histograms show means ± SD; *** *p* < 0.001, **** *p* < 0.0001. One way ANOVA followed by Tukey’s multiple comparisons test.

**Figure 10 pharmaceutics-14-02402-f010:**
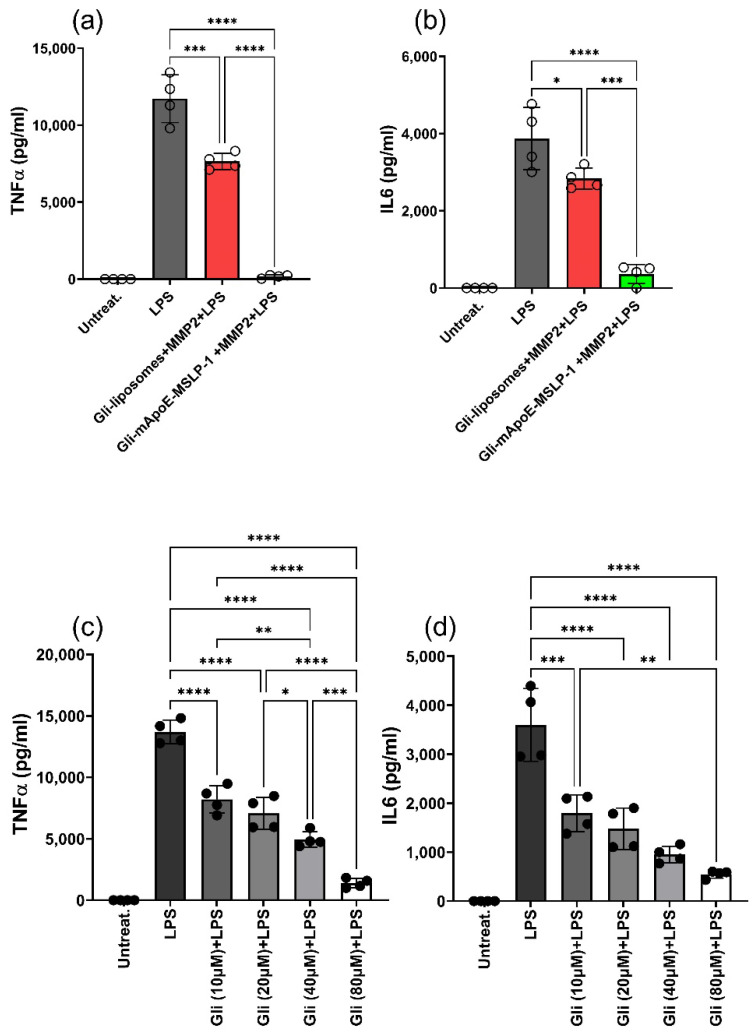
(**a**,**b**) BV2 cells (80 k/well) were exposed to activated MMP2, MMP-sensitive (green) and insensitive (red), glibenclamide (40 µm)-loaded mApoE liposomes for 60 h. Cells were then stimulated with LPS treatment for 8 h before assaying TNFα (**a**) and IL-6 (**b**) in supernatants; (**c**,**d**) BV2 cells (80 k/well) were exposed to increasing amounts of free glibenclamide (10–80µM) for 60 h before LPS stimulation for 8 h. The levels of both cytokines were established by an ELISA assay, dots indicate individual experiments, and histograms show means ± SD; * *p* < 0.05, ** *p* < 0.01, *** *p* < 0.001, **** *p* < 0.0001. One way ANOVA followed by Tukey’s multiple comparisons test.

## Data Availability

Not applicable.

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
