# Peer review of "Dual Functionalized Liposomes for Selective Delivery of Poorly Soluble Drugs to Inflamed Brain Regions"

_pharmaceutics, 2022, doi:10.3390/pharmaceutics14112402_

Round 1

Reviewer 1 Report

Acceptable format manuscript, however, authorshave to describe to limitation of the study.

Author Response

Answers to Reviewer’s #1 comments

  1. Acceptable format manuscript, however, author shave to describe to limitation of the study.

Thanks for the comment. The potential limitation of the study has been added in the discussion section (lanes 635-639, pp. 16-17).

Reviewer 2 Report

The work presented by Giofrè, et al focuses on the development of glibenclamide-encapsulated liposomes, functionalized with ApoE moieties (blood-brain-barrier targeting) and metalloproteinase sensitive lipopeptides (triggered release). The authors characterized the formulations and evaluated them in vitro, either with primary neuronal cultures or with microglia murine cells. Overall, the manuscript is well written and presents interesting results. Some aspects require further clarification, particularly regarding the experiments on neurons, before being considered for publication in Pharmaceutics.

Additional comments

I have some concerns regarding the experiments on neurons.

1.      In figure 7A (6 dots), 7B (5 dots) and 7C (3 dots), how many independent experiments were conducted? What does each dot represent?

2.      If they both arise from the same supernatant, why is the number of individual experiments different between 7B and 7C?

3.      What conclusion can be inferred from Figure 7A?

4.      What was the concentration of DMSO used in Figure 7B and 7C? It should be one that does not compromise neuron viability.

Calcein release from liposomes incubated with MMPs is rather low (<5%). Do authors have further information regarding release for longer time periods? This potential limitation should be addressed in the discussion, as it may limit the applicability of the liposomes in more complex and biologically relevant environments, not addressed in the study.

L34-L35 “(…) they led to a controlled release of proinflammatory cytokines from activated microglial cells.” Glibenclamide acts by reducing the release of proinflammatory cytokines.

L64 Replace “did” by “does”

L97 Verb is missing

L217-L223 Cumulative release assays either require independent samples per time points or a cumulative release equation, considering the mass losses due to sampling/media reposition.

was performed with independent. Are the time points independent?

L226-L227 Include a brief information regarding the MMP activation (with AMPA or other).

L254 Replace “nmols” by “nmol”

L156/L373 According to the methods, liposomes were produced by film hydration and sonication, not extrusion.

L375/Figure 3a Avoid using MSLP/Sm ratio, as it confuses the reader.

L406 Authors should include a table / edit Figure 3d, detailing the qualitative and quantitative composition of the formulations.

L424-L25 Replace “aspecific” by ”unspecific”.

L490 The information is also valid for the remaining lipopeptides. Authors should also clearly state that the key factor for crossing the BBB is the mApoE functionalization.

L535-L537 Reference missing.

L577 Correct accordingly

L659 Correct TNFα and IL-6 accordingly.

Overall, figure axis/legends difficult the reading and interpretation of the data, particularly, Figure 7, 9 and 10 when printed.

Author Response

Answers to Reviewer’s #2 comments

  1. In figure 7A (6 dots), 7B (5 dots) and 7C (3 dots), how many independent experiments were conducted? What does each dot represent? If they both arise from the same supernatant, why is the number of individual experiments different between 7B and 7C?

Dots showed in each panel of figure 7 indicate individual wells used for doing these measurements. Data included in panel A show the toxicity of DMSO in neuronal cultures that were exposed to increasing amounts of this solvent. These experiments have been repeated twice. In each replica we used 3 wells/condition. On the other hand, experiments in panels 7B and C show the toxicity of glibenclamide and glibenclamide-loaded mApoE-MSLP-1-liposomes (dose response). These experiments have been done on further preparations of neurons. We ran a first experiment on neurons treated with glibenclamide and with glibenclamide-loaded mApoE-MSLP-1-liposomes, using 2 wells per condition and we only measured CCK8 levels. We repeated this experiment on an additional batch of neurons treated with glibenclamide and with glibenclamide-loaded mApoE-MSLP-1-liposomes at the same concentrations used in the first experiment, using this time 3 wells/condition. In this second experiments we used part of cell supernatants to also determine LDH release. Data referring to CCK8 cytotoxicity were pulled and used to generate panel 7B, while data referring to LDH release were used to generate panel 7C.

  1. What conclusion can be inferred from Figure 7A?

We did not observe a clear dose/dependent toxicity in neurons treated with increasing amounts of DMSO. Even at the highest dosage we used, DMSO caused only a small reduction of CCK8 levels. These data were instrumental to evaluate glibenclamide-mediated cytotoxicity, as we show in panels 7B and 7C.

  1. What was the concentration of DMSO used in Figure 7B and 7C? It should be one that does not compromise neuron viability.

The highest concentration of DMSO used in experiments of figure 7b and c was 0.85%, that is tolerated by the cells as demonstrated in figure 7A.

  1. Calcein release from liposomes incubated with MMPs is rather low (<5%). Do authors have further information regarding release for longer time periods? This potential limitation should be addressed in the discussion, as it may limit the applicability of the liposomes in more complex and biologically relevant environments, not addressed in the study.

We agree with the reviewer comment. This point has been addressed in the discussion section (lanes 635-639, pp. 16-17).

  1. L34-L35 “(…) they led to a controlled release of proinflammatory cytokines from activated microglial cells.” Glibenclamide acts by reducing the release of proinflammatory cytokines. L64 Replace “did” by “does”. L97 Verb is missing. L226-L227 Include a brief information regarding the MMP activation (with AMPA or other). L254 Replace “nmols” by “nmol”. L375/Figure 3a Avoid using MSLP/Sm ratio, as it confuses the reader. L424-L25 Replace “aspecific” by ”unspecific”.

All these suggestions have been accepted, and the manuscript has been revised accordingly.

  1. L217-L223 Cumulative release assays either require independent samples per time points or a cumulative release equation, considering the mass losses due to sampling/media reposition was performed with independent. Are the time points independent?

Cumulative release assays were performed on independent aliquots derived from the same liposomal suspension. This concept has been clarified in the revised manuscript.

  1. L156/L373 According to the methods, liposomes were produced by film hydration and sonication, not extrusion.

Thanks for the suggestion. It was a typing mistake, which has been corrected in the revised version of the manuscript.

  1. L406 Authors should include a table / edit Figure 3d, detailing the qualitative and quantitative composition of the formulations.

The Table in Figure 3d has been implemented with the composition of the formulations in the revised manuscript.

  1. L490 The information is also valid for the remaining lipopeptides. Authors should also clearly state that the key factor for crossing the BBB is the mApoE functionalization.

We completely agree with the reviewer suggestion. Therefore, the key role of mApoE has been made more explicit in the revised manuscript.

10. L535-L537 Reference missing.

A reference has been added in the text.

11. L577 Correct accordingly.

 Corrected in the text.

12. L659 Correct TNFα and IL-6 accordingly.

 Corrected in the text.

13. Overall, figure axis/legends difficult the reading and interpretation of the data, particularly, Figure 7, 9 and 10 when printed.

Figures have been modified increasing the size and fonts.

Reviewer 3 Report

The manuscript “Dual-Functionalized Liposomes for Selective Delivery of Poorly Soluble Drugs to Inflamed Brain Regions” is devoted to the development of multifunctional nanocarriers with both a modified peptide derived from apolipoprotein E and the MMP-responsive lipopeptides. Such modified liposomes penetrate the BBB and release poorly soluble glibenclamide, which is poorly permeable to the CNS, in pathological, inflamed areas of the brain. The manuscript is a complete scientific work, reliable results have been obtained, from which reasonable conclusions follow. The materials and methods are written in sufficient detail to be reproducible.

However, there are the following several remarks to the article:

1 1.       It would be better to move the Figure 1 to the page 2 for to better visualize the purpose of the article.

22. P.2, L. 59. Please check here and below, all abbreviations are better deciphered.

33.   P.4, L.147. It would be better to split the section “2.3. Preparation and characterization of MSLP-liposomes functionalized with mApoE peptide” into several subsections according to the assays.

44.   P. 5, L. 201. It is not clearly written – what means “adding 9:1 isopropanol water (600 μL)”? 9 parts isopropanol and 1-part water?

55.   P.6, L. 258. It should be better to split the section to “Isolation, culture and treatment of primary neurons and cell lineages” and “The extraction total RNA”.  The same comments apply to “2.7. Cellular assays using MLSP-liposomes”.

66.     P. 7. L. 313. The sentence “The assay utilizes the colorless tetrazolium salt WST-8 [2-(2-meth- oxy-4-nitrophenyl)-3-(4-nitrophenyl)-5-(2,4-disulfophenyl)-2H-tetrazolium, monosodium salt] that is converted in orange formazan dye via cellular metabolism” is unnecessary, it is public knowledge.

77.   Figure 2, 3, 4, 6, 7, 9. Please, make the names of the molecules, liposomes and the labels of the axes in a larger font.

88.   P.8, L. 361. It is better not to use an abbreviation at the beginning of a sentence, please use circular dichroism instead.

99.  P.10. Figure 3. (e). Representative TEM image of 413 MSLP-liposomes – there is the problem with scale bar on the picture. Please, correct.

110.  P.12. Figure 6. (b). Please add the scale bar on the picture.

111. P. 13, L. 527. It should be better to explain the choice of Abcc8 transcripts as target for RT-PCR.

112.  P.17, L. 639. Please, avoid excessive use of abbreviations in the text, you have a lot of them, and this somewhat spoils the presentation of the content of the work (EP=endothelial permeability).

113.  P. 17, L. 659. There is a strange symbol after “TNF”.

114.   P.17. Ll. 642-652. Please describe here and in the Introduction, how the action of Sur1-KATP channels affects acute and chronic neuroinflammation, this is the goal setting of your work, why these targets were chosen, what is the molecular mechanism of action, etc.

Author Response

Answers to Reviewer’s #3 comments

  1. It would be better to move the Figure 1 to the page 2 for to better visualize the purpose of the article. P.2, L. 59. Please check here and below, all abbreviations are better deciphered. P.8, L. 361. It is better not to use an abbreviation at the beginning of a sentence, please use circular dichroism instead. P. 17, L. 659. There is a strange symbol after “TNF”. P.17, L. 639. Please, avoid excessive use of abbreviations in the text, you have a lot of them, and this somewhat spoils the presentation of the content of the work (EP=endothelial permeability).

These suggestions have been accepted and the manuscript has been revised accordingly.

  1. 4, L.147. It would be better to split the section “2.3. Preparation and characterization of MSLP-liposomes functionalized with mApoE peptide” into several subsections according to the assays.

The 2.3 section has been divided in two subsections, following the suggestion of the reviewer.

  1. 5, L. 201. It is not clearly written – what means “adding 9:1 isopropanol water (600 μL)”? 9 parts isopropanol and 1-part water?

Thanks for your comment. This point has been clarified in the revised manuscript.

  1. 6, L. 258. It should be better to split the section to “Isolation, culture and treatment of primary neurons and cell lineages” and “The extraction total RNA”. The same comments apply to “2.7. Cellular assays using MLSP-liposomes”.

We agree with the reviewer comment and these sections have been separated in the revised manuscript. Accordingly, the Methods sections have been renumbered.

  1. 7. L. 313. The sentence “The assay utilizes the colorless tetrazolium salt WST-8 [2-(2-meth- oxy-4-nitrophenyl)-3-(4-nitrophenyl)-5-(2,4-disulfophenyl)-2H-tetrazolium, monosodium salt] that is converted in orange formazan dye via cellular metabolism” is unnecessary, it is public knowledge.

This sentence has been deleted in the revised manuscript.

  1. Figure 2, 3, 4, 6, 7, 9. Please, make the names of the molecules, liposomes and the labels of the axes in a larger font.

Figures have been modified increasing the size and fonts.

  1. 10. Figure 3. (e). Representative TEM image of 413 MSLP-liposomes – there is the problem with scale bar on the picture. Please, correct. Figure 6. (b). Please add the scale bar on the picture

Scale bars of Fig 3e and Fig 6b have been modified in revised manuscript.

  1. 13, L. 527. It should be better to explain the choice of Abcc8 transcripts as target for RT-PCR.

 We have added the following sentence in the text: Abcc8 encodes for the ATP binding cassette (ABC) of Sur1, and therefore we wanted to assess whether BV2 cells used in our experiments were expressing this transcript.

  1. 17. Ll. 642-652. Please describe here and in the Introduction, how the action of Sur1-KATP channels affects acute and chronic neuroinflammation, this is the goal setting of your work, why these targets were chosen, what is the molecular mechanism of action, etc.

We inserted this short chapter in the introduction:

Glibenclamide efficiently inhibits the sulfonylurea receptor 1 (Sur1) [7], that assembles in both the Sur1-Trpm4 and Sur1-Kir6.2 ion channels [8]. Glibenclamide exerts protective effects in several CNS disorders including: subarachnoid haemorrhage, traumatic brain injury, ischemic stroke, and status epilepticus [9-13]. Furthermore, glibenclamide-mediated inhibition of Sur1 can dampen pro inflammatory mediators such as tumor necrosis factor (TNF) a, interleukin (IL)-6, and NF-kB in models mimicking cardiac arrest [14]. Because of this large body of literature indicating protective effects of glibenclamide, we selected it as a cargo to load our innovative nanoparticles

As a consequence, the references have been renumbered in revised manuscript.

Round 2

Reviewer 2 Report

The authors have answered my questions and followed my recommendations.